# PromptPilot: Game-Theoretic Multi-Agent Prompt Optimization for Segment Anything

Guangze Shi [* 1]   Yingjie Mi [* 1]   Jia Shen [1]   Feixue Shao [1]   Jiarui Cao [1]   Yexin Lai [1]   Xueyu Liu [1]   Rui Wang [1]
Yongfei Wu [1]   Mingqiang Wei [1 2]

## Abstract

Few shot segmentation with vision foundation models relies on high quality prompts to segment unseen categories from limited support annotations. Existing prompt construction methods depend on test-time adaptation, fixed heuristic sampling, or monolithic reinforcement learning, making it difficult to balance semantic consistency, spatial coverage, and prompt credit assignment. To address these limitations, PromptPilot is proposed as a hierarchical multi-agent reinforcement learning framework for point prompt optimization with frozen DINOv2 and SAM. Prompt construction is formulated as sequential decision making, where feature and physical agents propose complementary prompt modifications, and a manager agent selects actions using SAM feedback and local marginal contribution. PromptPilot functions as an inference-time optimization strategy without parameter updates. Extensive experiments demonstrate that the proposed game-theoretic approach improves segmentation performance and generalization, offering a principled solution for automated prompt engineering. The code is available at https://github.com/L-AILab/PromptPilot.

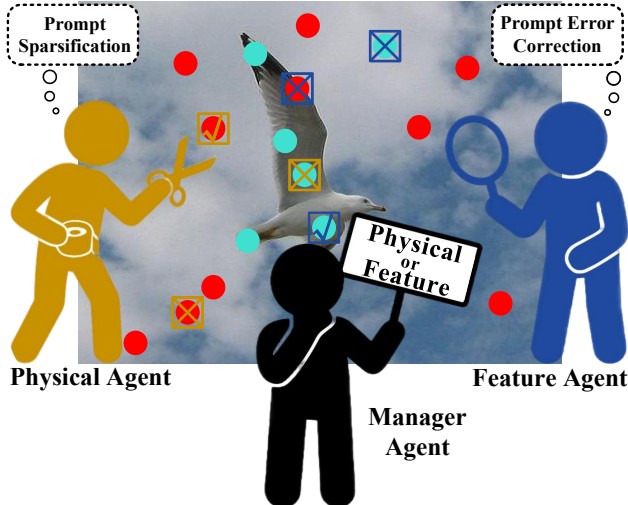

*Figure 1.* Conceptual illustration of PromptPilot. The Physical agent removes spatially redundant prompts to sparsify, while the Feature agent corrects semantically misaligned prompts. A centralized Manager agent adaptively chooses which agent to activate on the current prompt distribution.

## 1. Introduction

Image segmentation aims to identify target objects by predicting pixel level masks and plays a critical role in visual understanding. However, reliable segmentation models are usually trained with large scale annotations at the pixel level.

Such annotations are expensive to acquire, labor intensive to produce, and often impractical to obtain in specialized domains such as medical imaging, remote sensing, and industrial defect detection, since expert knowledge is usually required. To alleviate this annotation burden, few shot segmentation (FSS) has been studied to segment previously unseen categories with only a few annotated support examples. Traditionally, this field has been dominated by architecture engineering. In this paradigm, task specific networks are trained through episodic learning, and class consistent prototypes (Zhang et al., 2019; Wang et al., 2019) or dense pixel wise correlations (Min et al., 2021; Zhang et al., 2021) are learned to transfer category information from support images to query images. Although these fully trained models can achieve promising performance in closed set scenarios, their cross domain generalization remains limited, and costly adaptation is often required for novel tasks.

With the emergence of vision foundation models (VFMs), such as DINOv2 (Oquab et al., 2024) and the Segment

---

*Equal contribution   [1]Taiyuan University of Technology, Taiyuan, China [2]Nanjing University of Aeronautics and Astronautics, Nanjing, China. Correspondence to: Xueyu Liu (Lead Contact) <liuxueyu@tyut.edu.cn>, Yongfei Wu <wuyongfei@tyut.edu.cn>, Mingqiang Wei <mqwei@nuaa.edu.cn>.

*Proceedings of the 43rd International Conference on Machine Learning*, Seoul, South Korea. PMLR 306, 2026. Copyright 2026 by the author(s).

Anything Model (SAM) (Kirillov et al., 2023), a paradigm shift toward prompt engineering has occurred. Leveraging the zero-shot capability of SAM, recent studies have developed prompt construction approaches for few shot segmentation, including attention-based prompt generation, heuristic prompt selection, and reinforcement learning based prompt optimization. Attention-based methods, such as PerSAM (Zhang et al., 2024) and VRP-SAM (Sun et al., 2024), encode semantic correspondences via cross-attention or learnable prototypes. However, since these approaches often require test-time adaptation or task-specific encoder training, they undermine the task-agnostic efficiency of VFMs and hinder generalization to diverse unseen domains.

Heuristic-based strategies, represented by Matcher (Liu et al., 2024b), GBMSeg (Liu et al., 2024a) and SAT (Liu et al., 2025a), employ static rules for prompt selection, such as K-means prompt clustering in Matcher and exclusive, sparse, and hard negative sampling in GBMSeg or SAT. Although these methods are training-free, they rely on fixed heuristics that lack interactive feedback from the segmentation model, leading to suboptimal configurations unable to adapt to complex scenes. More recently, single agent reinforcement learning approaches, such as PPO (Liu et al., 2025b), have attempted to automate this process. Yet, because they rely on a monolithic policy that treats the prompt set holistically, these methods fail to decouple the conflicting objectives of spatial coverage and semantic consistency.

Consequently, three fundamental challenges inherent to high-dimensional prompt search are often overlooked by current optimization paradigms. First, an intrinsic conflict exists between optimization objectives. When semantic consistency is maximized, prompts tend to be concentrated in discriminative regions. In contrast, when spatial coverage is maximized, outlying prompts may be introduced and feature purity can be degraded. These competing objectives are difficult to optimize simultaneously under a monolithic policy. As a result, prompt configurations are often biased toward one objective, and suboptimal segmentation performance is consequently obtained. Second, a mismatch between the optimization criterion and the segmentation objective is introduced by existing methods. During optimization, prompt quality is assessed independently of SAM. However, during inference, segmentation performance is ultimately determined by the responses of SAM to the optimized prompts.

Third, prompt optimization is plagued by credit assignment ambiguity. Since the final segmentation mask is produced from a set of point prompts, the contribution of each individual prompt is coupled with those of the remaining prompts rather than being independently reflected in the prediction. As a result, the marginal utility of a specific prompt cannot be effectively isolated by standard approaches that rely only on global feedback. Inefficient policy updates are therefore

induced, since points with low utility may still be retained when favorable performance is achieved by the prompt set as a whole. The convergence toward a concise and robust prompt configuration is thus impeded.

To address these limitations, we propose PromptPilot, a hierarchical multi-agent reinforcement learning framework for prompt optimization. The overall concept is illustrated in Fig. 1. Prompt optimization is formulated as a cooperative game, where the optimization landscape is structurally decomposed into two complementary subspaces. Two specialized actor agents, namely the Feature agent and the Physical agent, are instantiated to optimize semantic discriminability and spatial diversity, respectively. To resolve conflicts between actor agents, a centralized Manager agent is employed as an adaptive arbiter, and actor actions are selected through a hybrid reward mechanism. More importantly, an efficient marginal contribution approximation is introduced to address the credit assignment problem in multi-agent sequential decision making. In this way, the marginal contribution of each prompt can be quantified by the Manager, and the policy can be guided toward optimal configurations that reconcile semantic purity with spatial coverage. Our contributions are summarized as follows:

- A hierarchical multi-agent reinforcement learning framework, termed PromptPilot, is proposed to formulate prompt optimization as a cooperative game, enabling efficient inference on unseen domains without parameter updates to the foundation model.

- The conflict between semantic consistency and spatial coverage is disentangled through a multi-agent architecture, where specialized actor agents are coordinated by a centralized Manager agent.

- Segmentation feedback of SAM is introduced as a global reward for Manager updates, aligning prompt optimization with the final segmentation objective.

- Credit assignment ambiguity in sequential decision making is addressed through a marginal contribution approximation, which quantifies each prompt utility to guide the policy toward high-utility configurations.

## 2. Method

As illustrated in Fig. 2, PromptPilot is developed for few shot segmentation with frozen DINOv2 and SAM, where point prompts for a target image are optimized from a reference image mask pair. Rather than training a task specific segmentor or updating the foundation model, prompt construction is formulated as sequential decision making over a target patch graph initialized by reference to target correspondences. The graph encodes complementary feature and physical relations, feature and physical agents propose

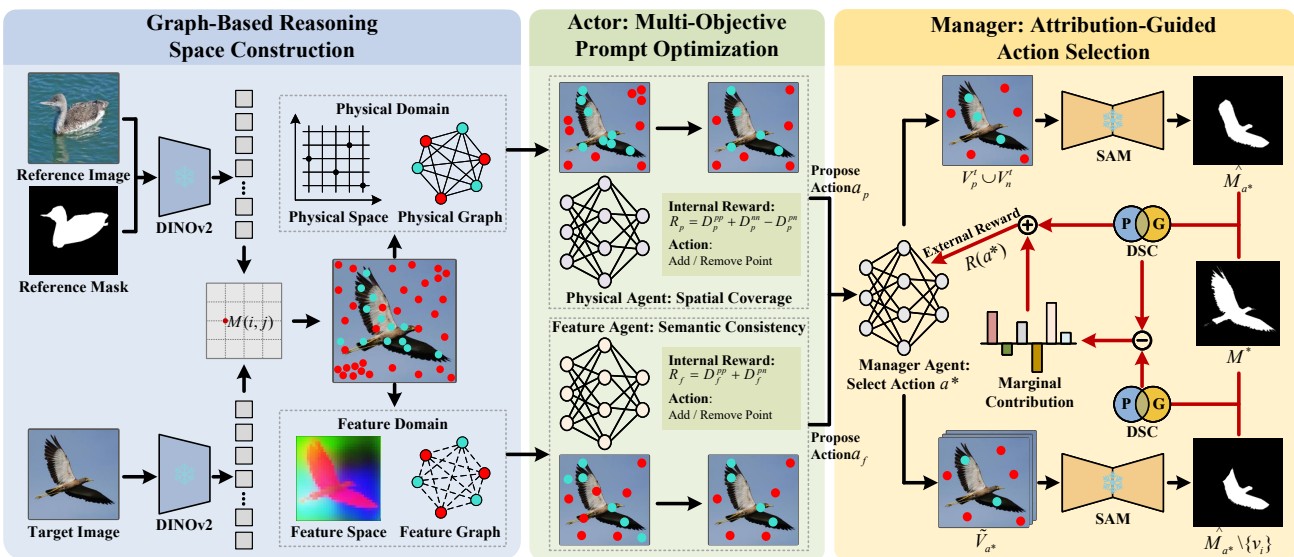

*Figure 2.* The workflow of PromptPilot is presented where the optimization landscape is decomposed into complementary physical and semantic domains. Candidate actions are independently proposed by the Physical and Feature agents to improve spatial coverage and feature consistency, respectively. The proposed actions are arbitrated by a centralized Manager agent, where a hybrid reward derived from global segmentation feedback and local marginal contribution approximations is used to execute the optimal policy update.

candidate prompt modifications under decoupled objectives, and a centralized manager agent selects the executed action based on SAM feedback and local marginal contribution.

### 2.1. Graph-Based Reasoning Space Construction

To instantiate the dual space environment, the reference image mask pair $(X_r, M_r)$ and the target image $X_t$ are uniformly discretized into grids of non-overlapping $14 \times 14$ patches. These patches form the reference patch set $X^r = \{x_i^r\}$ and the target patch set $X^t = \{x_j^t\}$, respectively. The spatial centroids of reference and target patches are denoted as $c_i^r \in \mathbb{R}^2$ and $c_j^t \in \mathbb{R}^2$.

Patch level feature embeddings $\{f_i^r\}$ and $\{f_j^t\}$ are extracted by a frozen DINOv2 encoder. The reference patch label $y_i^r \in \{0, 1\}$ is obtained from $M_r$, where foreground and background are denoted by 1 and 0, respectively. To establish a coarse prompt initialization, non-parametric label propagation is performed through bidirectional nearest neighbor matching by minimizing the Euclidean feature distance $\|f_i^r - f_j^t\|_2$ in the DINOv2 feature space.

For each reference patch $x_i^r$, forward matching is first performed over all target patches:

$$j^+(i) = \arg\min_j \|f_i^r - f_j^t\|_2. \qquad (1)$$

The label $y_i^r$ is assigned to the matched target patch $x_{j^+(i)}^t$ as a candidate pseudo label. For each target patch $x_j^t$ with a candidate pseudo label, reverse matching is then performed

to find its nearest reference patch:

$$i^-(j) = \arg\min_k \|f_k^r - f_j^t\|_2. \qquad (2)$$

The candidate pseudo label is retained only when the forward and reverse matching results provide consistent reference labels. Specifically, let $\mathcal{Y}_j = \{y_i^r \mid j^+(i) = j\}$ denote the candidate label set assigned to $x_j^t$ by forward matching. The retained pseudo label is defined as:

$$\hat{y}_j = \begin{cases} y_{i^-(j)}^r, & \text{if } y_{i^-(j)}^r \in \mathcal{Y}_j, \\ \varnothing, & \text{otherwise.} \end{cases} \qquad (3)$$

After label propagation, each target patch $x_j^t$ is represented as a node $v_j$ located at its centroid, and the node set is defined as $V = \{v_j\}$. The positive and negative prompt sets are initialized as two label specific subsets of $V$:

$$V_p = \{v_j \mid \hat{y}_j = 1\}, V_n = \{v_j \mid \hat{y}_j = 0\}, \qquad (4)$$

Target nodes without retained pseudo labels are excluded from the initial prompt sets.

The reasoning environment is formally defined as a dual relational graph $G = (V, E)$ over the target patches. Each node $v_i \in V$ corresponds to a target patch $x_i^t$ and is associated with its feature embedding $f_i^t$ and spatial centroid $c_i^t$. The edge set $E$ defines two complementary graph structures in the feature and physical spaces. Two distance matrices are computed over all node pairs $v_i, v_j \in V$ to characterize the graph structures in the feature and coordinate spaces:

$$M_f(i, j) = \|f_i^t - f_j^t\|_2, \qquad (5)$$

$$M_p(i,j) = \|c_i^t - c_j^t\|_2. \qquad (6)$$

Here, $M_f$ measures semantic dissimilarity in the embedding space, while $M_p$ measures spatial proximity in the image plane. This dual graph formulation provides the decoupled state representation required by subsequent agents to optimize semantic consistency and spatial coverage.

## 2.2. Actor: Multi-Objective Prompt Optimization

The optimization landscape is decomposed into complementary feature and physical spaces, which are represented by the dual graph structures constructed in Sec. 2.1. Two specialized actor agents are employed, namely the Feature agent $Q_{\theta_f}$ and the Physical agent $Q_{\theta_p}$. Given the current prompt state on the dual relational graph, candidate prompt modifications are generated to optimize prompts under decoupled intrinsic objectives. The Feature agent is guided to improve semantic discriminability, while the Physical agent is guided to improve spatial coverage.

**Action Space.** Both agents operate within a shared discrete action space $\mathcal{A}$, where $V$ is the target node set and $V_p, V_n \subseteq V$ are the positive and negative prompt node sets. At each timestep $t$, a discrete action $a_t \in \mathcal{A}$ is selected from two operations: (1) Activation, where an inactive node $v \in (V \setminus (V_p \cup V_n))$ is inserted into the prompt set to introduce additional semantic evidence; (2) Pruning, where an existing prompt node $v \in (V_p \cup V_n)$ is removed to reduce spatial redundancy or semantic noise. Validity constraints are imposed to ensure that the resulting prompt configuration can be accepted by the downstream foundation model.

**State Representation.** The environment state $s_t$ is defined as $s_t = (V_p, V_n, M_f, M_p)$, where $V_p$ and $V_n$ denote the current positive and negative prompt sets. The matrices $M_f$ and $M_p$ denote the feature distance matrix and the spatial distance matrix, respectively. This representation allows the Q-networks to observe both semantic relationships in the feature graph and spatial distributions in the physical graph.

**Reward Signal.** Dense intrinsic rewards are designed from the measurable geometry of the dual space. In the feature space, Euclidean distances between DINOv2 patch embeddings are used as a surrogate for semantic consistency, which is consistent with the patch matching criterion used in the initialization stage. In the physical space, Euclidean distances between prompt centroids on the image plane are used to characterize spatial coverage, redundancy, and foreground-background separation.

The feature agent optimizes semantic separability by reducing intra-class feature variation and enlarging inter-class feature margins. The feature reward $R_f$ is defined as:

$$D_f^{pp} = \frac{1}{|V_p|^2} \sum_{i,j \in V_p} M_f(i,j), \qquad (7)$$

$$D_f^{pn} = \frac{1}{|V_p||V_n|} \sum_{i \in V_p} \sum_{j \in V_n} M_f(i,j), \qquad (8)$$

$$R_f = -D_f^{pp} + D_f^{pn}, \qquad (9)$$

where $D_f^{pp}$ measures the compactness of positive prompts in the feature space, and $D_f^{pn}$ measures the separability between positive and negative prompts.

The physical agent optimizes spatial diversity and boundary awareness. The physical reward $R_p$ encourages same-class prompts to be spatially dispersed while keeping positive and negative prompts close enough to provide boundary cues:

$$D_p^{pp} = \frac{1}{|V_p|^2} \sum_{i,j \in V_p} M_p(i,j), \qquad (10)$$

$$D_p^{nn} = \frac{1}{|V_n|^2} \sum_{i,j \in V_n} M_p(i,j), \qquad (11)$$

$$D_p^{pn} = \frac{1}{|V_p||V_n|} \sum_{i \in V_p} \sum_{j \in V_n} M_p(i,j), \qquad (12)$$

$$R_p = (D_p^{pp} + D_p^{nn}) - D_p^{pn}. \qquad (13)$$

where $D_p^{pp}$ and $D_p^{nn}$ measure the spatial dispersion of positive and negative prompts, respectively, while $D_p^{pn}$ measures the spatial distance between the two prompt types.

**Policy Learning.** Policy optimization is performed by Deep Q-Learning (DQN). In the hierarchical framework, the global segmentation reward is available only for the action selected by the manager agent. Therefore, a conditional off-policy update scheme is used to couple the intrinsic objectives of actor agents with the global segmentation objective. Let $a^*$ denote the action executed at timestep $t$, which drives the transition from $s_t$ to $s_{t+1}$. Let $a_k$ denote the candidate action proposed by agent $k \in \{f, p\}$. The transition tuple $(s_t, a^*, s_{t+1})$ is shared by both actor agents, while different reward signals are used for policy learning. The reward for agent $k$ is defined as:

$$r_t^k = R_k(s_{t+1}) + \mathbb{I}(a^* = a_k) \cdot \lambda R(a^*), \qquad (14)$$

where $R_k(s_{t+1})$ denotes the intrinsic reward of agent $k$, $\mathbb{I}(\cdot)$ is the indicator function, $\lambda$ is a weighting coefficient, and $R(a^*)$ denotes the global hybrid reward. In this way, the unselected agent is updated by its intrinsic reward, while the selected agent is additionally guided by the global reward derived from segmentation feedback.

## 2.3. Manager: Attribution-Guided Action Selection

The manager agent serves as the centralized decision module for selecting between the actions proposed by the feature

---

**Algorithm 1** PromptPilot Training Process

---

**Input**: Graph environment $G = (V, E)$
**Parameter**: Max iteration $I$, Max steps $T$, Action space $\mathcal{A}$, Experience replay buffer $\mathcal{B}$, $k \in \{f, p\}$
**Output**: $Q_{\theta_f}, Q_{\theta_p}, Q_{\theta_m}$

1: Initialize $Q_{\theta_f}, Q_{\theta_p}, Q_{\theta_m}$ and $\mathcal{B}$
2: **for** $i = 1$ to $I$ **do**
3:    Initialize $V_p, V_n$ and set $s_0 = (V_p, V_n, M_f, M_p)$
4:    **for** $t = 1$ to $T$ **do**
5:       $a_f \leftarrow Q_{\theta_f}(s_t, \mathcal{A}), \quad a_p \leftarrow Q_{\theta_p}(s_t, \mathcal{A})$
6:       $a^* \leftarrow Q_{\theta_m}(\{a_f, a_p\})$
7:       Execute $a^*$ and update $V_p, V_n$ to obtain $s_{t+1}$
8:       $R_t \leftarrow R(a^*)$ by Eq. 19
9:       Compute $r_t^f$ and $r_t^p$ by Eq. 14
10:      Store $(s_t, a^*, s_{t+1}, R_t, r_t^f, r_t^p)$ in $\mathcal{B}$
11:      Sample a mini-batch from $\mathcal{B}$
12:      Update $Q_{\theta_m}$ with $R_t$
13:      Update $Q_{\theta_f}$ with $r_t^f$ and $Q_{\theta_p}$ with $r_t^p$
14:    **end for**
15: **end for**
16: **return** $Q_{\theta_f}, Q_{\theta_p}, Q_{\theta_m}$

---

and physical agents. Given the current state $s_t$, the two actor agents propose actions $a_f$ and $a_p$. The manager agent selects the action $a^* \in \{a_f, a_p\}$ according to $Q_{\theta_m}$. After execution, the resulting prompt configuration is evaluated by a hybrid reward derived from SAM segmentation feedback and local marginal contribution.

**Global Evaluation Metric.** For the executed action $a^*$, the resulting prompt set is obtained by applying $a^*$ to the current prompt set $V_p^t \cup V_n^t$. This operation is denoted as $\tilde{V}_{a^*} = (V_p^t \cup V_n^t) \oplus a^*$, where $\oplus$ indicates node insertion for activation or node removal for pruning. Given the target image $X_t$ and the candidate prompt set, the predicted mask is generated by SAM:

$$\hat{M}_{a^*} = \text{SAM}(X_t, \tilde{V}_{a^*}). \tag{15}$$

The segmentation quality is measured by the Dice similarity coefficient (DSC):

$$\text{DSC}(\tilde{V}_{a^*}) = \frac{2 \cdot |\hat{M}_{a^*} \cap M^*|}{|\hat{M}_{a^*}| + |M^*|}. \tag{16}$$

where $M^*$ denotes the ground truth mask. By introducing SAM feedback into this evaluation, prompt optimization is aligned with the final segmentation objective.

**Game-Theoretic Prompt Coalition.** However, relying only on DSC evaluates the prompt set as a whole and cannot identify whether a specific prompt is helpful, redundant, or harmful. To address this issue, prompt optimization is viewed as a finite cooperative game under a fixed image and graph state. Each prompt is treated as a player, the current prompt set $V_p^t \cup V_n^t$ forms a coalition, and the DSC score produced by SAM defines the coalition value. Under this view, credit assignment requires estimating the marginal contribution of each prompt to the current coalition.

Several strategies can be used for marginal contribution estimation. For example, the exact Shapley value computation estimates each contribution to the prompt by enumerating all possible subsets of the prompt and provides attribution based on the principle of coalition, but requires $\mathcal{O}(2^{|V_p^t \cup V_n^t|})$ SAM evaluations at each decision step, which is infeasible for sequential decision making online. Sampling based estimation, such as random subset evaluation, reduces the number of evaluated coalitions, but the induced variance may destabilize online policy learning. Perturbation based estimation, such as prompt masking or coordinate perturbation, is computationally efficient, but the estimated score may be unreliable when prompt interactions are strong. Since the optimization process follows sequential decision making, the manager agent mainly requires the local utility of a prompt under the current coalition, rather than its global contribution over all possible coalitions. Therefore, leave one out (LOO) is adopted as an efficient local marginal contribution estimator, and EMA is applied to smooth temporal fluctuations in the estimated contribution.

**LOO Marginal Contribution.** Local marginal contribution of each prompt $v_i \in \tilde{V}_{a_k}$ is defined as:

$$\delta_i = \text{DSC}(\tilde{V}_{a^*}) - \text{DSC}(\tilde{V}_{a^*} \setminus \{v_i\}). \tag{17}$$

A positive $\delta_i$ indicates that removing $v_i$ decreases the segmentation quality, so the prompt is beneficial to the evaluated coalition. A negative $\delta_i$ indicates that removing $v_i$ improves the segmentation quality, so the prompt is harmful under the current state. The sign of $\delta_i$ is determined by its effect on SAM segmentation quality and is independent of whether $v_i$ belongs to the positive or negative prompt set.

To reduce temporal fluctuation, an exponential moving average (EMA) is used to update the prompt importance score:

$$w_i^{(t)} = \alpha w_i^{(t-1)} + (1-\alpha)\delta_i, \tag{18}$$

where $\alpha \in [0, 1]$ is the momentum coefficient. This score records the historical signed contribution of each prompt and provides local attribution for manager decision making.

**Hybrid Reward Scalarization.** The manager agent selects the executed action by evaluating the hybrid reward of each candidate action:

$$R(a^*) = \beta \cdot \text{DSC}(\tilde{V}_{a^*}) + (1-\beta) \cdot \frac{1}{|\tilde{V}_{a^*}|} \sum_{v_i \in \tilde{V}_{a^*}} w_i^{(t)}. \tag{19}$$

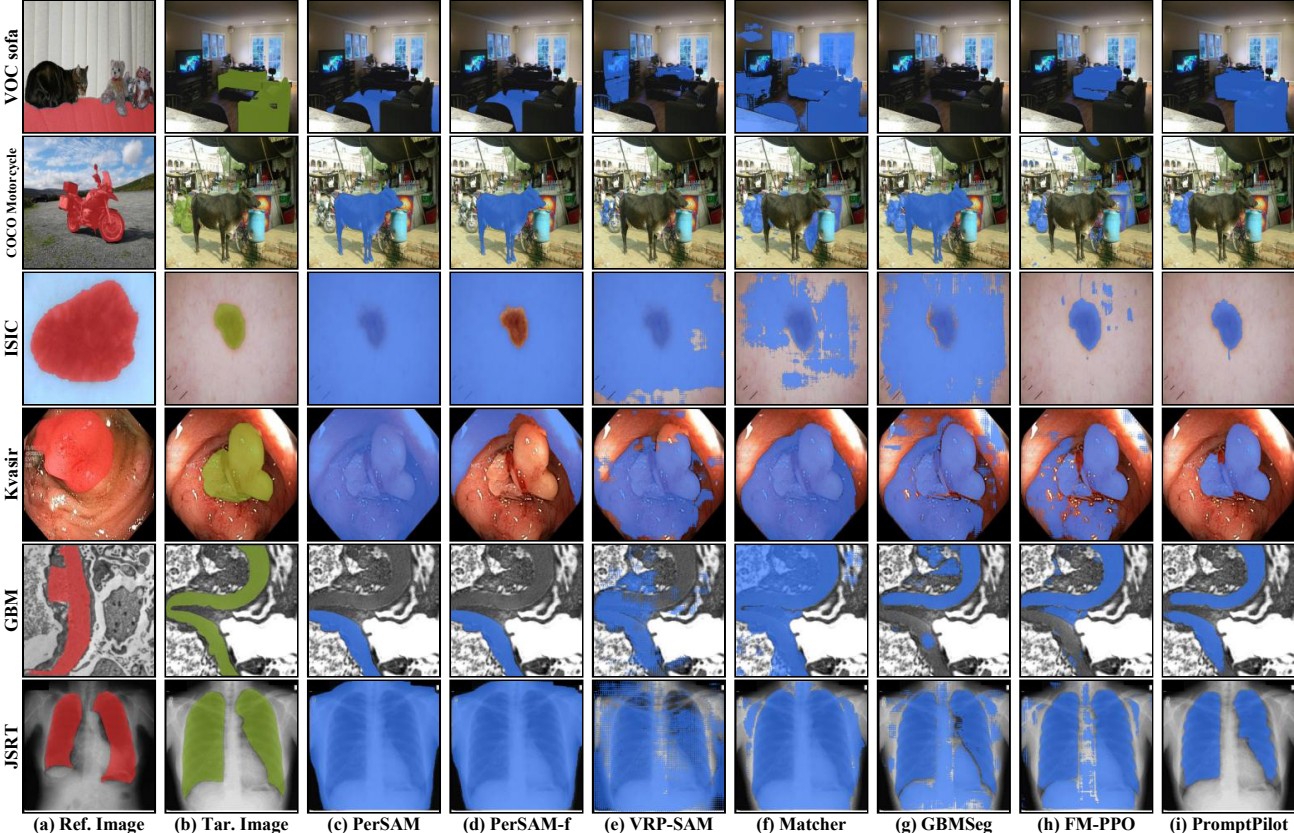

*Figure 3.* Qualitative comparison of segmentation results across two natural benchmarks, VOC and COCO, and four medical datasets including ISIC, Kvasir, GBM, and JSRT.

where $\beta \in [0, 1]$ controls the balance between global segmentation quality and local prompt attribution. In this way, hybrid reward is used to update the manager agent and to provide the global feedback for the selected actor agent.

## 3. Experiments

### 3.1. Dataset Description

Our method is evaluated on eight datasets covering natural image segmentation, medical image segmentation, and video object segmentation. For natural images, PASCAL VOC (Everingham et al., 2010) and COCO (Lin et al., 2014) are used, which contain 20 and 80 object categories, respectively. For fair comparison, 100 test images are randomly selected from each category of these benchmarks. For medical images, ISIC2018 Task 1 (Codella et al., 2018) is used for skin lesion boundary segmentation, whose official test split contains 1,000 dermoscopic images used in this study, and Kvasir (Jha et al., 2020) provides 1,000 labeled endoscopic images of gastrointestinal lesions. JSRT (Shiraishi et al., 2000) contains 60 chest X-ray images with annotated lung boundaries, while GBM (Liu et al., 2024a) includes 184 transmission electron microscopy images of glomerular

basement membranes manually labeled by clinical experts. For video object segmentation, DAVIS 2016 (Perazzi et al., 2016) focuses on single object sequences, while DAVIS 2017 (Pont-Tuset et al., 2017) introduces multi object scenarios with higher spatial and temporal complexity.

### 3.2. Experimental Settings

All experiments are conducted on an Ubuntu 22.04 server with an NVIDIA GeForce RTX 4090 GPU. During training, the maximum number of iterations is set to $I = 200$, and each episode contains $T = 100$ steps. The EMA momentum coefficient $\alpha = 0.9$, balancing coefficient $\beta = 0.5$, and reward weighting coefficient $\lambda = 0.1$ are adopted. The feature matching paradigm (Liu et al., 2024b;a; 2025b) is used to initialize prompts, where a single annotated reference image provides initial guidance for the target image. During inference, the learned agents optimize the prompt configuration without parameter updates, and the final mask is generated by the frozen SAM backbone. Performance is evaluated under a unified protocol using DSC and mean Intersection over Union (mIoU) for static image segmentation, while Region Similarity ($\mathcal{J}$) and Contour Accuracy ($\mathcal{F}$) are reported for video object segmentation.

*Table 1.* Comparison of segmentation performance on both medical and natural image datasets. All results are rounded to one decimal place for consistency. The best results are highlighted in **bold**, and the second-best results are underlined. The last row ($\Delta$) denotes the performance gain of PromptPilot over the second-best method. Rows in gray indicate that manual prompts are provided for each image.

| Method | VOC | | COCO | | ISIC | | Kvasir | | GBM | | JSRT | |
|---|---|---|---|---|---|---|---|---|---|---|---|---|
| | DSC | mIoU | DSC | mIoU | DSC | mIoU | DSC | mIoU | DSC | mIoU | DSC | mIoU |
| SAM (Point) (Kirillov et al., 2023) | 50.4 | 41.2 | 52.2 | 42.8 | 69.3 | 60.1 | 72.4 | 63.6 | 65.4 | 52.5 | 59.6 | 43.1 |
| SAM (Box) (Kirillov et al., 2023) | 80.1 | 72.8 | 72.3 | 64.5 | 84.6 | 75.9 | 78.6 | 72.0 | 55.8 | 41.7 | 60.8 | 45.9 |
| SAM-Adapter (Chen et al., 2023) | – | – | – | – | 74.7 | 65.4 | 86.3 | 79.1 | 51.9 | 41.6 | 90.0 | 82.2 |
| SAM-LoRA | – | – | – | – | 91.0 | 84.4 | 83.4 | 75.8 | 67.5 | 55.5 | 87.3 | 81.6 |
| PerSAM (Zhang et al., 2024) | 55.3 | 49.3 | 26.5 | 22.7 | 63.9 | 54.5 | 29.3 | 19.9 | 43.6 | 33.2 | 58.2 | 41.5 |
| PerSAM-f (Zhang et al., 2024) | 54.5 | 48.0 | 23.8 | 20.2 | 60.6 | 51.4 | 33.3 | 25.0 | 45.2 | 35.2 | 57.3 | 40.8 |
| VRP-SAM (Sun et al., 2024) | 48.8 | 40.8 | 25.2 | 19.2 | 64.5 | 55.5 | 28.0 | 17.9 | 19.7 | 12.3 | 49.8 | 33.4 |
| Matcher (Liu et al., 2024b) | 68.9 | 59.9 | 45.0 | 38.4 | 74.7 | 66.3 | 39.4 | 29.5 | 58.8 | 45.3 | 89.5 | 81.3 |
| GBMSeg (Liu et al., 2024a) | 55.3 | 47.7 | 32.9 | 26.9 | 59.2 | 48.3 | 40.2 | 28.9 | 59.1 | 45.0 | 79.6 | 66.9 |
| FM-PPO (Liu et al., 2025b) | 61.3 | 52.8 | 36.7 | 30.4 | 72.4 | 62.7 | 44.9 | 33.9 | 66.0 | 53.1 | 87.7 | 78.4 |
| **PromptPilot (Ours)** | **69.3** | **61.3** | **54.4** | **47.1** | **78.6** | **69.0** | **49.3** | **40.1** | **72.5** | **60.6** | **89.8** | **81.7** |
| $\Delta$ | +0.4 | +1.4 | +9.4 | +8.7 | +3.9 | +2.7 | +4.4 | +6.2 | +6.5 | +7.5 | +0.3 | +0.4 |

*Table 2.* Comparison results on DAVIS video object segmentation benchmarks. The best results are highlighted in **bold**, and the second-best results are underlined.

| Method | DAVIS2016 | | DAVIS2017 | |
|---|---|---|---|---|
| | $\mathcal{J}$ | $\mathcal{F}$ | $\mathcal{J}$ | $\mathcal{F}$ |
| PerSAM(Zhang et al., 2024) | 68.2 | 69.7 | 61.7 | 67.7 |
| PerSAM-f(Zhang et al., 2024) | 69.3 | 70.6 | 64.9 | 70.3 |
| VRP-SAM(Sun et al., 2024) | 47.4 | 47.4 | 35.1 | 39.0 |
| Matcher(Liu et al., 2024b) | 78.5 | 81.7 | 65.1 | **73.0** |
| GBMSeg(Liu et al., 2024a) | 76.5 | 74.6 | 56.1 | 54.9 |
| FM-PPO(Liu et al., 2025b) | 75.9 | 76.0 | 61.5 | 62.4 |
| **PromptPilot (Ours)** | **80.1** | **81.8** | **69.3** | 71.1 |

### 3.3. Comparison with State-of-the-Art Methods

**Natural Image Segmentation.** The performance on natural image segmentation is evaluated on PASCAL VOC and COCO, as shown in Table 1. Interactive strategies, including SAM-Point and SAM-Box, are reported as manual prompt performance upper references, since their prompts are derived from the target masks. However, their reliance on human intervention limits their applicability to scalable automated inference. In the automated setting, the proposed method achieves mIoU scores of 61.3% on VOC and 47.1% on COCO, establishing new state-of-the-art performance among automated methods. These results indicate that the proposed framework substantially narrows the gap to manual prompt segmentation quality while requiring no human annotation during inference.

Substantial gains are observed over representative automated baselines. On COCO, the proposed method improves the mIoU by 8.7 percentage points over Matcher, showing that static clustering based prompt selection is insufficient when the final mask is determined by SAM responses.

Compared with the monolithic reinforcement learning baseline FM-PPO, gains of 8.5 and 16.7 percentage points are achieved on VOC and COCO, respectively. These results indicate that single policy optimization tends to entangle semantic consistency and spatial coverage, while the proposed multi-agent design provides more targeted prompt optimization through specialized feature and physical agents.

Qualitative comparisons in Fig. 3 further support the quantitative results. On VOC, the proposed method produces clearer object boundaries and preserves thin structures more effectively, while attention-based methods such as PerSAM and VRP-SAM often miss these details due to limited adaptation to large intra-class variation. In cluttered COCO scenes, Matcher and FM-PPO tend to activate background regions or produce over-segmented masks. In contrast, the proposed method suppresses irrelevant regions and maintains better semantic consistency. These results show that feedback-driven prompt optimization is effective for handling complex scenes with diverse object layouts, where heuristic selection and attention-based prompt construction are prone to performance degradation.

**Cross-Domain Medical Segmentation.** Cross-domain generalization is evaluated on ISIC, Kvasir, GBM, and JSRT, as summarized in Table 1. Interactive or adapted SAM variants, including SAM-LoRA and SAM-Adapter, are reported as manual prompt performance upper references, since they benefit from high-quality manual prompts or domain-specific adaptation. In contrast, the proposed method performs automated prompt optimization without parameter updates during inference. On GBM, the proposed method achieves 60.6% mIoU, surpassing SAM-LoRA with 55.5% mIoU despite the domain-specific parameter updates used by the latter. On JSRT, an mIoU of 81.7% is obtained,

*Table 3.* Ablation study of different components in PromptPilot. The baseline uses a monolithic single-agent policy to jointly optimize semantic and spatial objectives. Best results are **bolded**, and second-best are underlined.

| Actor Agents | | Manager Agent | | ISIC | | Kvasir | | GBM | | JSRT | | Average | |
|---|---|---|---|---|---|---|---|---|---|---|---|---|---|
| Feature | Physical | DSC | LOO | DSC | mIoU | DSC | mIoU | DSC | mIoU | DSC | mIoU | DSC | mIoU |
| Single Agent | | | | | | | | | | | | | |
| ✓ | | | | 51.4 | 41.1 | 22.4 | 15.8 | 40.9 | 31.4 | 63.9 | 47.5 | 44.7 | 34.0 |
| | ✓ | | | 52.3 | 42.2 | 22.6 | 15.9 | 39.9 | 30.5 | 65.5 | 49.2 | 45.1 | 34.5 |
| ✓ | ✓ | | | 50.3 | 39.9 | 23.2 | 16.2 | 41.4 | 31.6 | 65.6 | 49.2 | 45.1 | 34.2 |
| Multi Agents | | | | | | | | | | | | | |
| ✓ | | ✓ | | 74.4 | 65.3 | 36.3 | 27.4 | 61.1 | 48.5 | 79.3 | 67.7 | 62.8 | 52.2 |
| | ✓ | ✓ | | 74.8 | 65.7 | 36.8 | 27.8 | 64.0 | 51.4 | 79.2 | 67.0 | 63.7 | 53.0 |
| ✓ | ✓ | ✓ | | 77.9 | 68.3 | 47.7 | 38.4 | 71.1 | 58.9 | 88.3 | 79.6 | 71.3 | 61.3 |
| ✓ | ✓ | ✓ | ✓ | **78.6** | **69.0** | **49.3** | **40.1** | **72.5** | **60.6** | **89.8** | **81.7** | **72.6** | **62.9** |

which is comparable to the manual prompt reference. These results indicate that SAM feedback guided prompt optimization can approach manual prompt performance in medical segmentation without human intervention during inference.

The results also reveal the limitations of existing prompt construction strategies under domain shifts. VRP-SAM degrades to 12.3% mIoU on GBM, suggesting that attention-driven correspondence may be unreliable when the target modality differs significantly from natural images. Compared with feedback-free heuristic baselines, the proposed method achieves a gain of 7.5 percentage points over the second-best FM-PPO on GBM. This improvement shows that prompt quality in medical images cannot be sufficiently determined by feature or spatial heuristics alone. By introducing SAM segmentation feedback and local marginal contribution into the manager update, prompt optimization is better aligned with the final mask quality, which is especially important for low-contrast medical structures.

Qualitative results further confirm these observations. In electron microscopy images from GBM, the proposed method separates thin membrane structures more accurately, while FM-PPO and GBMSeg often produce fragmented or incomplete masks. On ISIC and Kvasir, more complete lesion regions are obtained from a single reference image, reducing the under-segmentation commonly observed in Per-SAM. These visual results demonstrate that the proposed framework provides robust automated prompt optimization across different medical modalities.

**Video Object Segmentation.** Temporal generalization is evaluated on the DAVIS benchmarks, as reported in Table 2. On DAVIS 2016, the proposed method achieves 80.1% region similarity $\mathcal{J}$ and 81.8% contour accuracy $\mathcal{F}$, outperforming the heuristic baseline Matcher. On DAVIS 2017, which contains more complex multi object scenarios, the proposed method obtains 69.3% $\mathcal{J}$ and surpasses

Matcher with 65.1% $\mathcal{J}$. This gain indicates that fixed heuristic prompt selection is limited under occlusion and object interactions. Prototype based methods such as VRP-SAM degrade to 35.1% $\mathcal{J}$, since static first frame features are difficult to generalize across temporal appearance changes. In contrast, the proposed method maintains more stable performance by optimizing prompts dynamically, which helps reduce error accumulation across video frames.

### 3.4. Ablation Study

An ablation study is conducted on four medical benchmarks to validate the contribution of each component, as summarized in Table 3. First, single-agent variants are evaluated to examine the effect of isolated optimization objectives. When only the feature objective is used, an average mIoU of 34.0% is obtained, indicating that semantic consistency alone tends to concentrate prompts in local discriminative regions and provides insufficient spatial coverage. When only the physical objective is used, the average mIoU slightly increases to 34.5%, but the absence of semantic regularization may introduce background or noisy prompts. When a single DQN agent jointly optimizes both objectives, the average mIoU remains limited at 34.2%. These results show that a monolithic policy is insufficient to balance semantic consistency and spatial coverage.

Clear improvements are achieved when global DSC feedback is introduced through the manager agent. Compared with the single-agent variants without manager supervision and features-only and physical-only, the corresponding variants supervised by a manager improve the average mIoU from 34.0% to 52.2% and from 34.5% to 53.0%, respectively. These gains indicate that SAM based global feedback is effective for reducing the mismatch between intrinsic optimization objectives and the final segmentation objective. When both actor agents are further coordinated by the manager agent, the average mIoU increases to 61.3% without

*Table 4.* Comparative analysis of segmentation performance across four medical datasets is presented. The generalization capability of PromptPilot is evaluated against baselines including Feature Matching and Coarse Segmentation of UNet (Ronneberger et al., 2015), with the best results highlighted in **bold**.

| Setting | Method | ISIC | | Kvasir | | GBM | | JSRT | | Average | |
|---|---|---|---|---|---|---|---|---|---|---|---|
| | | DSC | mIoU | DSC | mIoU | DSC | mIoU | DSC | mIoU | DSC | mIoU |
| **Feature Matching** | Baseline | 71.6 | 61.5 | 36.9 | 28.8 | 49.9 | 38.9 | 73.2 | 60.5 | 57.9 | 47.4 |
| | FM-PPO | 72.4 | 62.7 | 44.9 | 33.9 | 66.0 | 53.1 | 87.7 | 78.4 | 67.8 | 57.0 |
| | **FM-PromptPilot** | **78.6** | **69.0** | **49.3** | **40.1** | **72.5** | **60.6** | **89.8** | **81.7** | **72.6** | **62.9** |
| **Coarse Segmentation** | Baseline | 46.0 | 36.0 | 20.4 | 14.5 | 25.4 | 18.3 | 44.9 | 31.2 | 34.2 | 25.0 |
| | CS-PPO | 51.2 | 41.1 | 22.4 | 16.2 | 21.2 | 14.7 | 58.5 | 42.3 | 38.3 | 28.6 |
| | **CS-PromptPilot** | **78.5** | **69.0** | **45.9** | **36.4** | **71.2** | **58.4** | **89.3** | **81.1** | **71.2** | **61.2** |

LOO marginal contribution. This result shows that global segmentation feedback provides effective task-level guidance, while decomposing prompt optimization into feature and physical spaces further alleviates the conflict between semantic consistency and spatial coverage.

The full configuration achieves the best performance, with an average DSC of 72.6% and an average mIoU of 62.9%. Compared with the manager variant using only DSC feedback, the LOO marginal contribution further improves the average mIoU by 1.6 percentage points. This gain indicates that global segmentation feedback alone is insufficient for identifying the utility of individual prompts. By incorporating local marginal contribution with EMA smoothing, the manager agent can assign credit more effectively and guide prompt optimization toward more reliable configurations.

### 3.5. Plug-and-Play Study

The plug-and-play ability of PromptPilot is evaluated under two initialization settings, as summarized in Table 4. In the feature matching setting, the baseline achieves an average mIoU of 47.4%, and FM-PPO improves it to 57.0%, indicating that reinforcement learning can optimize prompts initialized from feature correspondences. However, the gain remains limited by a monolithic policy that couples semantic consistency and spatial coverage, and by prompt objectives that are not directly aligned with SAM responses. With the same initial prompts, FM-PromptPilot further increases the average mIoU to 62.9%. This gain indicates that the proposed framework improves segmentation performance by decomposing the conflicting objectives into feature and physical spaces and using SAM feedback to align optimization with the final segmentation objective.

A larger performance gap is observed in the coarse segmentation setting, where the initial pseudo labels are generated by a UNet trained with only 5% of the dataset. The coarse segmentation baseline obtains 25.0% average mIoU, and CS-PPO only improves it to 28.6%. This result suggests that single-agent optimization is sensitive to inaccurate initial prompts, since semantic errors, spatial redundancy, and

background drift must be corrected within a unified policy. In contrast, CS-PromptPilot achieves 61.2% average mIoU and 71.2% average DSC. The improvement shows that PromptPilot can perform more effective prompt optimization under arbitrary initial prompts.

## 4. Conclusion

In few shot segmentation, vision foundation models rely on high quality prompts to segment unseen target categories from limited annotated support examples. However, prompt construction is not a static selection problem, since semantic consistency and spatial coverage must be balanced while the contribution of each prompt remains coupled with the current prompt set. Therefore, prompt optimization is formulated as a high dimensional sequential decision process, where semantic consistency, spatial coverage, and credit assignment are jointly considered. To address these challenges, PromptPilot is proposed as a hierarchical multi-agent reinforcement learning framework for prompt optimization. The optimization space is decomposed into complementary feature and physical spaces, where specialized feature and physical agents optimize semantic discriminability and spatial coverage, respectively. A centralized manager agent is further introduced to coordinate the actor agents through SAM segmentation feedback and local marginal contribution. By using LOO with EMA smoothing, prompt utility can be estimated under the current coalition, enabling more reliable credit assignment for sequential decision making.

Extensive experiments on natural image, medical image, and video object segmentation benchmarks consistently demonstrate the effectiveness of the proposed framework. Compared with heuristic, attention-based, and single-agent reinforcement learning baselines, PromptPilot achieves stronger generalization through coordinated multi-agent decision making under domain shifts and complex object interactions. These results show that feedback-driven prompt optimization can better align the optimization process with the final segmentation objective, while requiring no parameter updates to the foundation model during inference.

## Acknowledgements

This work was supported by the National Natural Science Foundation of China (No. T2322012, No. 62572240, No. 62572339), the Fundamental Research Program of Shanxi Province (No. 202303021211082), and Key Research and Development Program of Shanxi Province (No. 202402020101008).

## Impact Statement

This work studies prompt optimization for few shot segmentation with vision foundation models. The proposed hierarchical multi-agent reinforcement learning framework improves the coordination between semantic consistency and spatial coverage, and reduces the dependence on manual prompt design during inference. Such a framework may be useful for annotation-efficient segmentation in domains where pixel level labels are expensive to obtain, including medical image analysis, remote sensing, and industrial defect detection. In medical scenarios, the proposed method may support computer-aided analysis and dataset annotation by reducing the amount of expert interaction required for prompt generation. However, this work remains a research-stage study. All evaluations are conducted on publicly available benchmark datasets under controlled experimental settings. The results are not intended to support direct clinical decision making or to replace human experts in safety-critical applications. All experiments are performed on publicly released datasets. These datasets are anonymized and were collected under the ethical protocols of the original data providers. No additional private patient data are collected or used in this study.

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
