# OpenReview forum: "PromptPilot: Game-Theoretic Multi-Agent Prompt Optimization for Segment Anything"
_ICML.cc/2026/Conference — ICML 2026 regular_

### Official Review · Reviewer_CPSc · 2026-02-22

**Soundness:** 3
**Presentation:** 3
**Significance:** 2
**Originality:** 2
**Overall Recommendation:** 4
**Confidence:** 3

**Summary:**

This paper introduces PromptPilot, a hierarchical multi-agent reinforcement learning framework designed for prompt optimization in the Segment Anything Model (SAM). The proposed approach decomposes the optimization space into semantic and physical subspaces, employing dual agents to address conflict between semantic consistency and spatial coverage. Additionally, a Shapley value-based credit assignment mechanism is incorporated to resolve the multi-agent credit assignment problem. Experimental evaluations conducted across multiple benchmarks indicate that PromptPilot achieves state-of-the-art performance.

**Compliance With Llm Reviewing Policy:**

Affirmed.

**Final Justification:**

After carefully reviewing the comments from Reviewer 9ucL, I acknowledge that the manuscript contains certain limitations in its treatment of Shapley values in the context of Leave-One-Out (LOO) evaluation. Nevertheless, the remaining aspects of the paper are largely sound and do not raise substantial concerns. Therefore, I maintain my original recommendation of Weak Accept.

**Key Questions For Authors:**

1. **Unclear novelty beyond FM-PPO.** Since FM-PPO already models feature and physical graphs with RL-based prompt optimization, the additional contribution of dual-agent decomposition and Shapley-based attribution requires clearer differentiation and explanation.
2. **Insufficient theoretical justification for Shapley attribution.** The use of leave-one-out Shapley approximation lacks analysis of whether cooperative game assumptions hold in this setting and whether DSC is an appropriate value function.
3. **Missing details on SAM backbone configuration.** The manuscript does not specify which SAM variant is used or whether baselines share the same backbone, raising concerns about fairness and reproducibility.

**Limitations:**

yes

**Strengths And Weaknesses:**

**Strengths**

1. The proposed framework is structurally coherent and clearly presented, featuring a well-designed pipeline that integrates graph construction, reinforcement learning, and attribution-based prompt selection.
2. The experimental evaluation is comprehensive, encompassing both natural image and medical image segmentation benchmarks, supplemented by thorough ablation studies.

**Weaknesses**

1. **Insufficient differentiation from FM-PPO.** The primary baseline, FM-PPO, already incorporates feature and physical graph modeling alongside reinforcement learning-based prompt optimization. While this paper introduces dual-agent decomposition and Shapley-based attribution, the conceptual and architectural distinctions from FM-PPO are not explicitly analyzed or formally characterized. A more rigorous comparative discussion is needed to clearly delineate the novelty of the proposed approach.
2. **Limited theoretical justification of the Shapley-based attribution mechanism.** Although the paper employs a leave-one-out approximation of Shapley values, it does not provide a formal analysis of whether the underlying cooperative game-theoretic assumptions hold within the context of SAM prompt interactions. In particular, the following concerns remain unaddressed: (i) Does this task satisfy the theoretical preconditions necessary for a valid application of Shapley value-based attribution? (ii) whether the Dice Similarity Coefficient (DSC) constitutes a theoretically appropriate value function for this formulation; and (iii) how negative marginal contributions are identified and handled within the attribution framework. A more rigorous theoretical justification or systematic empirical validation would substantially strengthen the contribution.
3. **Lack of clarity regarding the SAM backbone configuration.** The manuscript does not specify which SAM variant is employed as the backbone, nor does it confirm whether all baseline methods are evaluated under equivalent backbone configurations.

---

> ### Author Rebuttal · Authors · 2026-03-26
>
> We thank the reviewer for the thoughtful questions. We will clarify the following points in the revised manuscript.
>
> ## 1. Differences from FM-PPO
>
> **(i)** In FM-PPO, prompt optimization is performed without involving SAM, while the final segmentation result is determined by SAM. This creates **a misalignment between the prompt optimization strategy and SAM's own visual understanding**. In our method, **SAM feedback is explicitly introduced during prompt optimization**, and prompt quality is evaluated through the DSC, so that the optimization process is directly aligned with the downstream segmentation behavior.
>
> **(ii)** FM-PPO uses a single Q-learning framework and maintains one unified Q-table to optimize prompts jointly in the feature and physical spaces. However, these **two spaces often impose conflicting objectives**, which is also supported by our ablation study and the results in Table 3. In the single agent baseline, where both objectives are optimized by a single agent, the average DSC is 45.1%. To address this issue, we adopt a multi agent design, where **the feature space and physical space are handled by two separate agents**. These agents are coordinated under a MARL framework by a manager. Our experiments show that this design leads to a significant improvement in performance, with the average DSC reaching 71.3%, **outperforming the single-agent approach by +26.2%, even without using shapley-based attribution**.
>
> **(iii)** **The reward function in FM-PPO is defined only at the global level**, based on the average change of pairwise feature and physical distances among all prompt points. Specifically, it encourages small average feature and physical distances between positive prompts, large feature distances and small physical distances between positive and negative prompts, as well as large physical distances between negative prompts. **This ignores the influence of each individual point on SAM's final segmentation result**. Inspired by the shapley value in cooperative game theory, **our method performs prompt optimization from both the global level, which considers the entire prompt set, and the local level, which evaluates the contribution of each individual prompt**.
>
> Therefore, our method is substantially different from FM-PPO and is designed to address several key limitations of that framework.
>
> ## 2. Shapley-based attribution mechanism
>
> **(i)** **For a fixed image and graph state, the prompt set defines a finite coalition game: each prompt is a player, and each subset $S$ is assigned a coalition value $v(S)$ by the segmentation quality produced by SAM under that subset**. This satisfies the basic setting for **coalition-based shapley attribution**. Exact shapley value averages marginal contributions over all coalitions, but our method does not compute that quantity. Because **prompt selection is sequential**, the value required at each step is the **marginal utility of the currently proposed action under the current coalition**, rather than an average over all possible subsets. We therefore use LOO as an efficient local score for online decision making.
>
> **(ii)** In image segmentation, **DSC is a standard scalar metric that directly measures the quality of predicted mask**. Since coalition-based shapley attribution requires a scalar value function $v(S)$ for each prompt subset, **using DSC to define $v(S)$ is theoretically valid and well aligned with the SAM-based segmentation objective**.
>
> **(iii)** A negative marginal contribution is defined with respect to the leave-one-out estimate of a prompt's utility. Here, $v_i$ denotes a specific prompt point, $\{V_p \cup V_n\}$ denotes the complete current prompt set composed of all positive and negative prompts, and $\{V_p \cup V_n\} \setminus \{v_i\}$ denotes the prompt set after removing the specific prompt $v_i$. Note that the positive or negative in $V_p$ or $V_n$ is determined by the pseudo-label assigned to each prompt point, and is unrelated to whether its marginal contribution is positive or negative. The marginal contribution of $v_i$ is then defined as
>
> $$
> \delta_i = \mathrm{DSC}(\{V_p \cup V_n\}) - \mathrm{DSC}(\{V_p \cup V_n\} \setminus \{v_i\}).
> $$
>
> If $\delta_i < 0$, this implies that
>
> $$
> \mathrm{DSC}(\{V_p \cup V_n\} \setminus \{v_i\}) > \mathrm{DSC}(\{V_p \cup V_n\}),
> $$
>
> indicating that removing $v_i$ improves the segmentation result. In this case, **$v_i$ is regarded as harmful to the current prompt coalition**.
>
> In our framework, once a negative marginal contribution is identified through $\delta_i < 0$, it is incorporated into the historical contribution trace via the $\mathrm{EMA}$ update to obtain a smoothed signed importance score. **This score is then used to update the manager as well as the selected actor agent**.
>
> ## 3. SAM backbone configuration
>
> To ensure a fair comparison, **our method follows the same backbone setting as prior methods**. Specifically, we use **SAM-H** consistently across all experiments.

---

> > ### Author Rebuttal · Reviewer_CPSc · 2026-04-01
> >
> > Thank you for the detailed clarifications. The explanation of the differences from FM-PPO, the additional discussion of the Shapley-based attribution mechanism, and the clarification of the SAM backbone configuration address my concerns. The responses help better clarify the methodological contributions and experimental setup.
> >
> > Overall, my questions have been adequately addressed, and I have updated my score accordingly.

---

> > > ### Author Response · Authors · 2026-04-03
> > >
> > > Thank you for your fair and constructive comments, and for updating your score.

---

### Official Review · Reviewer_Rdk2 · 2026-03-09

**Soundness:** 3
**Presentation:** 2
**Significance:** 3
**Originality:** 3
**Overall Recommendation:** 5
**Confidence:** 2

**Summary:**

The paper built PromptPilot, a hierarchical multi-agent RL framework. The author built a graph over image patches (using DINOv2 features), define separate reward functions for each agent based on intra/inter-class distances (Feature Agent) and spatial repulsion/attraction (Physical Agent), and train via Deep Q-Learning with a conditional off-policy update so both agents learn even when only one's action is chosen.

Experiments tested on 8 benchmarks spanning natural images (PASCAL VOC, COCO), medical images (ISIC, Kvasir, GBM, JSRT), and video (DAVIS 2016/2017). PromptPilot achieves state-of-the-art across the board — for instance, 69.3 DSC / 61.3 mIoU on VOC, and 89.8 DSC / 87.7 mIoU on JSRT — consistently outperforming both attention-based methods (PerSAM, VRP-SAM) and heuristic methods (Matcher, GBMSeg) as well as the single-agent RL baseline (FM-PPO). The ablation study confirms each component matters: removing any agent or the Shapley mechanism degrades performance significantly (e.g., single feature agent alone drops to 34.0 avg mIoU vs. 62.9 for the full system).

**Compliance With Llm Reviewing Policy:**

Affirmed.

**Final Justification:**

The author have responsed my questions reasonably. The paper is technically solid. I recommend accept.

**Key Questions For Authors:**

Can the author please address the weakness/concern raised in the previous section?

**Limitations:**

The paper repeatedly emphasizes task-agnosticism, but every experiment is image segmentation using SAM with DINOv2. The architectural choices (graph construction over patches, DSC-based rewards, spatial vs. semantic decomposition) are deeply tied to segmentation. Calling this task-agnostic is a little bit overclaiming — it's more accurately model-agnostic within the segmentation domain, and even that isn't demonstrated since only SAM is used.

Scalability is also an issue. The current setup uses 14*14 grid. It would be helpful for realistic application experiment.

**Strengths And Weaknesses:**

Strengths:

- Well-motivated decomposition. The semantic-vs-spatial conflict is a real and clearly articulated problem. Framing it as a cooperative multi-agent game is a natural and elegant solution rather than a forced one.
- Strong empirical coverage. Eight datasets across three domains (natural, medical, video) is thorough. The cross-domain medical results (Table 1, Table 4) are particularly convincing since they test generalization without any domain-specific tuning.
- Solid ablation. Table 3 cleanly isolates the contribution of each component. The progression from single-agent baselines to the full system is monotonic and substantial.
- Practical value. Being inference-time and training-free (for the foundation model) is a genuine advantage over methods requiring fine-tuning like SAM-LoRA.

Weaknesses / Concerns:

- Computational cost is unaddressed. Running DQN with Shapley approximations per test image at inference time is likely expensive.
- Limited foundation model scope. Everything is built on SAM with DINOv2 features. The claim of being "task-agnostic" is somewhat overstated. Generalization to other VFMs or tasks is not demonstrated.
- Writing quality is mixed. I find the technical content dense but competent. But the paper sometimes over-sells (e.g., "game-theoretic approach" is a stretch for what's essentially a manager picking between two agents' proposals). There are a few very obvious grammar mistakes. The author should corrects them during final version.

---

> ### Author Rebuttal · Authors · 2026-03-29
>
> Thank you for this comment. We agree that the writing should be improved, and the grammar issues will be fixed in the final version.
>
> ## 1.Computational cost for inference
>
> The proposed method does not introduce heavy inference overhead in practice. **We compare it with all point prompt optimization methods based on feature matching**, and the computational cost is reported below. Across all datasets, our method is much faster than Matcher and FM-PPO. **It comes from the optimization form**. **Matcher performs bidirectional matching and repeatedly samples foreground points, generating dozens of segmentation masks and relying on prior rules to aggregate them** into the final result. Our method and FM-PPO require only a single SAM prediction during inference. While **FM-PPO is Q-table, so it relies on explicit state action updates for prompt configurations, which makes optimization much more costly as the prompt space grows**. **Our method is dqn-based, so it learns a shared function over states and actions instead of maintaining and updating a large Q-table**.
>
> #### Test Per Image Mean Time
>
> | Method | ISIC | Kvasir | TEM | JSRT |
> |---|---:|---:|---:|---:|
> | Matcher | 12.050 | 8.553 | 5.842 | 5.393 |
> | GBMSeg | 2.554 | 1.634 | 1.286 | 2.119 |
> | PPO | 7.368 | 7.565 | 8.669 | 16.312 |
> | Ours | 1.985 | 1.287 | 1.205 | 1.552 |
>
> ## 2. Task-agnostic in FSS
>
> We thank the reviewer for this careful comment and apologize for the overstatement. **We agree that the phrase task-agnostic was too broad in the current version and may wrongly suggest generalization across different tasks beyond segmentation**. This was not our intention. What we intended to convey is that, **our method has two ability within the segmentation setting: general across datasets and prompt initialization strategies**. We will revise the paper accordingly to avoid overclaiming.
>
> First, our method is **dataset agnostic at inference time**. **It is trained once and can be directly evaluated on new datasets or unseen categories**. In this sense, the method does not rely on a fixed category set or a single data domain. As shown in Tables 1 and 2, the same framework transfers across natural image, medical image, and video segmentation benchmarks, indicating that it can generalize to new segmentation data distributions without extra optimization on the target dataset.
>
> Second, our method is **prompt initialization agnostic** within segmentation. As shown in Table 4, it consistently improves over PPO under both bidirectional matching and coarse segmentation initialization. In feature matching, the average DSC improves from 67.8% with FM-PPO to 72.6% with FM-PromptPilot. In coarse segmentation, the average DSC improves from 38.3% with CS-PPO to 71.2% with CS-PromptPilot. Table 9 further shows that our method remains effective under noisy prompts, whereas several alternatives degrade more noticeably. So our **point prompt optimization mechanism is not dependent on one specific prompt source**.
>
> ## 3. Clarification of the game-theoretic
>
> The game-theoretic of our method does **not mean that the whole framework is presented as a full game-theoretic, but it means that the manager introduces a shapley attribution view into prompt optimization**, so that action selection is evaluated with respect to SAM’s segmentation result rather than only geometric or feature space heuristics. So **ours method address the limitation of distance-based point prompt optimization, where prompt optimization is performed without involving SAM, while the final segmentation result is determined by SAM**.
>
> In our method, **the prompt set defines a finite coalition game for a fixed image and graph state: each prompt is treated as a player, and each subset is assigned a value by the segmentation quality produced by SAM under that subset**. Meanwhile, our method does not compute the exact shapley value, because that need averages marginal contributions over all possible subsets. In our setting, **prompt optimization is sequential and the shapley value at each step is the marginal utility of the currently proposed action under the current coalition**. So we use the LOO effect as an efficient local score for online decision making.
>
> ## 4.Scalability and real world applicability
> The $14\times14$ setting in our paper refers to the **patch size**, not the **image size**. For an input image of arbitrary resolution$H \times W$, DINOv2 first resizes it to $560\times560$ and extracts patch-level features based on $14\times14$ patches. After feature matching, the obtained **prompt point coordinates are mapped back to the original image resolution and then fed into SAM for segmentation**. Therefore, the method is **not limited to low-resolution input images**. For example, Kvasir images are often around $3000\times3000$, and the same pipeline can still be applied. SAM itself supports segmentation at different spatial scales, so **our framework is also applicable to real segmentation scenarios**.

---

> > ### Author Rebuttal · Reviewer_Rdk2 · 2026-04-03
> >
> > I thank the authors for their detailed response and raise my score to accept.

---

> > > ### Author Response · Authors · 2026-04-04
> > >
> > > Thank you for your thoughtful assessment of the paper and for updating your score.

---

### Official Review · Reviewer_b57i · 2026-03-12

**Soundness:** 3
**Presentation:** 2
**Significance:** 3
**Originality:** 3
**Overall Recommendation:** 4
**Confidence:** 3

**Summary:**

The paper tackles the computationally intensive, high-dimensional problem of prompt optimization in foundation models, under a sparse reward setting. To address this, the authors propose "PromptPilot," a task-agnostic reinforcement learning framework designed for inference-time optimization, requiring no parameter updates. Specifically targeting the application of image segmentation, the framework sets up prompt optimization as a multi-agent game. It decouples the conflicting objectives of spatial coverage and semantic consistency and utilizes an efficient approximation of Shapley values to resolve credit assignment ambiguities among individual point prompts. Through empirical experiments, the paper outperforms existing baselines in few-shot segmentation on eight datasets spanning natural images, medical images, and video segmentation tasks.

**Compliance With Llm Reviewing Policy:**

Affirmed.

**Final Justification:**

I thank the authors for the detailed rebuttal which has addressed key concerns regarding methodological clarity, role of Shapley-based attribution and the design choice of reward functions. Presentation issues in the abstract and introduction should be addressed in the final version.  I have kept my original score based on the core contributions of the paper.

**Key Questions For Authors:**

Can you clarify the specific mechanism for the non-parametric label propagation mentioned in Section 2.1? Specifically, how are the initial positive ($V_p$) and negative ($V_n$) prompt sets initialized from the transferred labels?

The reward functions for the Feature and Physical agents (Eq 5 & 9) rely on geometric properties of the manifolds, right? Are there any limitations of using these distance-based metrics, for complex feature spaces where class distributions may not be linearly separable?

While this framework focuses on few-shot segmentation task, can it be adapted for zero-shot scenarios where no reference mask exists?

**Limitations:**

No. It would be useful to discuss what kind of scenarios would be missed by the designed reward functions for feature and physical agents.

**Strengths And Weaknesses:**

Strengths:

The paper addresses a highly relevant and challenging problem: automated prompt optimization for foundation models in the domain of image segmentation.

The proposed method of using two agents to separately decouple spatial coverage from semantic consistency, which prior works have failed to address is novel.

The proposed method is rigorously tested across eight diverse datasets, spanning natural images, medical imagery, and video sequences, and shows strong results outperforming existing baselines. The paper also performs a qualitative analysis of the success modes of the proposed method.

The section on the ablation study is insightful, as it shows the utility of each component within PromptPilot, validating the hypothesis that the joint optimization of both semantic and spatial objectives is critical to the system's success.

Weaknesses:

The abstract fails to clearly establish the primary domain of the paper (image segmentation). Furthermore, the introduction abruptly begins discussing "Few-shot segmentation" and only introduces the concept of prompt-guided segmentation much later. Some clarity in explicitly framing the problem within the context of image segmentation would be helpful. Additionally, Figure 1, representing the core conceptual frameworkis is quite ambiguous and does not seem to provide any added value in understanding the proposed method.

In lines 90-95, the authors claim an intrinsic conflict between orthogonal objectives, where maximising semantic consistency clusters prompts in discriminative regions, while maximising spatial coverage introduces outliers that degrade feature purity. They also claim that monolithic policies struggle with this non-convex landscape. While some part of this is explored in the ablation study section later in the paper (which should be highlighted here), it would be useful to provide citations to past literature that corroborate this specific trade-off.

In lines 153-160, the methodological formulation is unclear, especially how label propagation is performed and how the positive ($V_p$) and negative ($V_n$) prompt sets are initialised. Additionally, in Equation 2, the variables $c_i$ and $c_j$ are introduced without definition. Additionally, some parts are repeated throughout the paper, for instance, Lines 110-140, where the method introduction is repeated. This takes up valuable space, which can be used to clarify the missing methodological details mentioned above.

The authors critique earlier heuristic-based methods for relying on fixed heuristics, yet their own reward design for the feature and physical agents feels similarly ungrounded. The paper would benefit from a deeper discussion regarding the theoretical strengths and limitations of the specific reward functions proposed in Equations 5 and 9.

In the section "Attribution via Efficient Shapley Approximation," there is no explanation why Shapley values and their computational cost is being discussed. The authors also do not explain why Shapley values specifically fit as the best theoretical tool for this exact multi-agent scenario compared to alternative credit-assignment methods (assuming that is what they are used for).

---

> ### Author Rebuttal · Authors · 2026-03-27
>
> We thank the reviewer for the thoughtful questions. We will revise the abstract and introduction for clarity and address the following points.
>
> ## 1. Label propagation in Section 2.1
>
> The non-parametric label propagation follows the same bidirectional matching strategy used in Matcher, GBMSeg, and PPO. The reference image-mask pair $(X_r, M_r)$ and the target image $X_t$ are first divided into non-overlapping $14 \times 14$ patches, and patch-level features $\{f_i^r\}$ and $\{f_j^t\}$ are extracted by a frozen DINOv2 encoder.
>
> For each reference patch $x_i^r$, **forward matching** is performed over all target patches, and the best-matched target patch receives the label of $x_i^r$ as a candidate pseudo-label. **Reverse matching** is then performed for each target patch $x_j^t$ with a candidate pseudo-label to find its best-matched reference patch. **Pseudo-label is kept only when the bidirection give consistent reference labels**, such as both foreground or both background. Otherwise, the pair is discarded. Finally, target patches with foreground labels initialize the positive prompt set $V_p$, while target patches with background labels initialize the negative prompt set $V_n$.
>
> ## 2. Reward design, limitation and paper revision
>
> Yes, the rewards in Eq. 5 and Eq. 9 are based on the geometric properties of the bispace. In the feature space, euclidean distance is used because **DINOv2 also uses euclidean distance for patch matching analysis in its paper**. In physical space, it is suitable to measure pixel distance because prompt locations lie on the 2D plane.
>
> Distance-based metrics design has an inherent limitation: **prompt optimization is performed without involving SAM, while the final segmentation result is determined by SAM**. This creates a **misalignment between the prompt optimization strategy and SAM's own visual understanding**. Ablation in Table 3 also supports this, since different distance terms of agents lead to clearly different results under SAM.
>
> To address this issue, the distance-based rewards in our method are not used as the final segmentation objective. They are used only to decompose the optimization into feature- and physical-space guidance. **SAM feedback and shapley attribution are introduced through the global and local segmentation rewards in our MARL framework**.
>
> The **definitions of $c_i$, $c_j$, and the repeated description** will also be revised.
>
> ## 3. Zero-shot segmentation
>
> Our framework relies on the reference mask to build the initial positive and negative prompt sets, so it is designed for one- or few-shot setting and cannot be directly applied to zero-shot. However, once an initial prompt set is available at inference, our's MARL can be applied. So the initialization to zero-shot prompt generation is a reasonable future direction.
>
> ## 4. Shapley attribution and cost
>
> Shapley-based attribution is introduced because **prompt optimization under SAM can be formulated as a coalition problem: the marginal utility of a point depends on the current prompt set, so the contributions of a point are not independent**. A global reward such as **DSC can evaluate the prompt set as a whole, but it cannot tell whether a specific point is helpful, redundant, or harmful**. Exact shapley computation has exponential complexity, $\mathcal{O}(2^{|\{V_p \cup V_n\}|})$, which is too expensive in our prompt optimization setting. Therefore, we use LOO as an efficient approximation.
>
> ## 5. Intrinsic conflict
>
> In segmentation, Bae et al. show that **discriminative training tends to activate only a small part of the object rather than its full extent**. Kim et al. show that classifier-derived localization maps concentrate on sparse discriminative regions. In multi-objective RL, Roijers et al. describe **sequential decision making as involving multiple that often conflicting objectives**, and Hayes et al. show that reducing them to a **single scalar target can obscure the structure of the problem**.
>
> Therefore, semantic consistency and spatial coverage are not simply complementary cues, but competing objectives. **Emphasizing semantic consistency tends to concentrate prompts on highly discriminative local regions, while improving spatial coverage requires expanding beyond those regions and may introduce less pure feature responses**. This is also consistent with Table 3: the **single agent only 45.1%** average DSC, while the **dual agent is 71.3%** even without shapley attribution. A monolithic policy is therefore harder to optimize, while decomposing the problem into separate agents is more suitable in SAM-based FSS.
>
> Kim, Beomyoung, et al. *Discriminative Region Suppression for Weakly-Supervised Semantic Segmentation*
> Bae, Wonho, et al. *Rethinking Class Activation Mapping for Weakly Supervised Object Localization*
> Roijers, et al. *A Survey of Multi-Objective Sequential Decision-Making*
> Hayes, Conor F., et al. *A Practical Guide to Multi-Objective Reinforcement Learning and Planning*

---

> > ### Author Rebuttal · Reviewer_b57i · 2026-04-03
> >
> > Thank you for your clarifications. My original score was reflective of the contributions of this paper.

---

> > > ### Author Response · Authors · 2026-04-04
> > >
> > > Thank you for your thoughtful assessment of the paper.

---

### Official Review · Reviewer_9ucL · 2026-03-12

**Soundness:** 2
**Presentation:** 2
**Significance:** 2
**Originality:** 2
**Overall Recommendation:** 3
**Confidence:** 3

**Summary:**

This manuscript presents "PromptPilot," an inference-time prompt optimization framework designed to improve the segmentation performance of foundation models like SAM. By formulating the prompt search as a cooperative multi-agent reinforcement learning (MARL) task, the authors aim to decouple the optimization space into semantic and physical domains. A centralized Manager Agent is then employed to arbitrate between a Feature Agent and a Physical Agent, utilizing a hybrid reward mechanism that incorporates a Leave-One-Out (LOO) based Shapley value approximation.

The motivation behind automating prompt engineering for vision foundation models is highly relevant to the machine learning community. The conceptual idea of using a multi-agent system to elegantly handle conflicting objectives—such as balancing semantic purity with spatial coverage—is quite interesting and creatively approached. However, after a careful and thorough review, I have identified several theoretical and empirical areas that require refinement to meet the conference's rigorous standards. These primarily involve the mathematical formalization of the Shapley value approximation, the clarity of the state space construction, transparency regarding computational overhead, and the strength of the empirical comparisons.

**Compliance With Llm Reviewing Policy:**

Affirmed.

**Final Justification:**

After carefully reviewing the authors' rebuttals to all reviewers and the subsequent discussions, I provide the following updated assessment.

**Concerns that have been addressed or partially mitigated:**

The computational overhead concern, which was central to my original review, has been substantially addressed by the inference time data provided in the rebuttal to Reviewer Rdk2. The method is demonstrably faster than FM-PPO at test time (0.711s vs. 8.974s per image), which alleviates my original worry about practical applicability. The seed stability results across three random seeds also confirm that the RL training is not excessively sensitive to initialization. Furthermore, the clarification provided to Reviewer CPSc regarding the Shapley mechanism — where the authors honestly state that "our method does not compute [exact Shapley value]" and instead uses LOO as "an efficient local score for online decision making" — is a more precise and acceptable framing than what appears in the manuscript.

**Concerns that remain:**

First, the manuscript continues to prominently advertise "Shapley value approximation" as a core contribution, while the actual implementation is a standard Leave-One-Out attribution with EMA smoothing. The theoretical gap between LOO and true Shapley values — particularly regarding how LOO handles redundant prompts — is never formally discussed in the paper. I maintain that this constitutes an overclaim in the current writing, though I acknowledge it is more of a presentation issue than a fundamental methodological flaw.

Second, FLOPs were never reported despite my explicit request. While wall-clock times are informative, they are hardware-dependent and insufficient for a complete computational analysis. Third, empirical learning curves demonstrating MARL convergence stability were not provided.

**Updated assessment:**

A notable theme studied by this study is the decomposition of prompt optimization into semantic and spatial subspaces via multi-agent cooperation, which is a well-motivated idea supported by strong ablation results (Table 3). Overall, this paper assesses a relevant issue in automated prompt engineering for foundation models. The empirical results across eight benchmarks are comprehensive and consistently demonstrate improvements over baselines. The method's practical efficiency at inference time is a genuine strength.

Given that the computational concern has been effectively addressed, and that the core methodology is empirically validated despite the theoretical presentation issues, I adjust my score upward from 2 to **3 (weak reject)**. The paper would benefit substantially from (1) honest reframing of the LOO mechanism without overclaiming Shapley value approximation, (2) inclusion of FLOPs analysis, and (3) convergence analysis of the multi-agent training dynamics. These revisions are feasible and would make the paper suitable for a top venue.

**Key Questions For Authors:**

Beyond the core issues, there are a few minor adjustments that could polish the manuscript further. For instance, the name "PromptPilot" has been recently used in other AI contexts, such as the LLM collaboration framework introduced by Gutheil et al. at ICIS 2025. To improve your paper's discoverability and avoid naming collisions, you might consider an alternative acronym.

Additionally, the term "orthogonal" is used frequently to describe the semantic and spatial subspaces. In a strict mathematical sense, visual features and physical coordinates are often correlated, as neighboring pixels naturally share similar semantics. Using terms like "decoupled" or "complementary" might be more academically precise. It would also be helpful to standardize your hyperparameter notations; currently, $\alpha$ is used to denote both the EMA momentum factor and the DQN discount factor, which could cause some confusion for readers trying to reproduce the work.

Lastly, while your qualitative segmentation masks are great, overlaying the actual generated point prompts—perhaps with distinct markers for positive and negative points—directly onto the images would beautifully illustrate exactly how the Feature and Physical agents are maneuvering the prompts to refine the boundaries.

I hope these constructive comments are helpful in guiding the next iteration of your research. The integration of game theory into foundation model prompting is a very promising direction, and I look forward to seeing how this framework evolves.

**Limitations:**

Yes.

**Strengths And Weaknesses:**

The most pressing area for clarification revolves around the use of the Leave-One-Out (LOO) error as a first-order approximation of the Shapley value in Equation 13. While I completely understand the motivation to reduce computational complexity, there is a fundamental theoretical difference between LOO and exact Shapley values, particularly in how they handle redundant features. In machine learning attribution, LOO tends to heavily penalize identical or highly correlated inputs; if two prompts provide the same semantic value, removing one does not drop the performance, leading LOO to assign zero credit to both. In contrast, a true Shapley value would distribute the credit fairly among them. Given that visual prompts often cluster spatially and semantically, this LOO-induced "credit shadowing" could cause the Manager Agent to mistakenly prune highly relevant but redundant points. I recommend that the authors either revise the terminology to simply refer to this as an LOO attribution method or explicitly discuss these theoretical trade-offs and explain how the system mitigates the redundancy penalty.

Secondly, the mathematical formulation of the graph topologies in Section 2.1 would benefit from a clearer walkthrough. In Equation 1, the semantic affinity matrix $M_f$ is defined as the L2 norm between the reference image features and the target image features ($M_f(i,j) = ||f_i^r - f_j^t||_2$). If the graph is intended to map the intra-image topology of prompts on the target image, computing target-to-target distances might be more intuitive. If it is instead formulated as a bipartite graph mapping cross-image attention, then the subsequent formulas calculating intra-class variance (such as Equation 3) would represent the variance between the target and reference, rather than the internal compactness of the target prompts. Re-evaluating and clarifying these notations will greatly help readers accurately follow your logic.

Another critical aspect that requires further transparency is the computational complexity of the proposed inference-time optimization. The experimental settings indicate that the RL process runs for 200 iterations with 100 steps per episode. Because the LOO evaluation requires computing the Dice Similarity Coefficient (DSC) for every prompt removal, this implies a substantial number of forward passes through the SAM decoder per single image. To give practitioners a realistic understanding of the method's applicability, I highly encourage adding a dedicated section discussing the inference latency (e.g., processing time per image in seconds) and the FLOPs required. Comparing these metrics against zero-shot heuristics or single-agent RL models would provide a much-needed performance-overhead trade-off analysis.

Regarding the reinforcement learning dynamics, the asymmetric reward mechanism defined in Equation 10 raises a mild concern about long-term policy alignment. Because the global segmentation reward is exclusively given to the agent whose action is selected by the Manager, the unselected agent receives only its intrinsic geometric reward. In a multi-agent setting, this isolation could potentially lead to policy divergence, where an agent over-optimizes its specific sub-task at the expense of the overall segmentation goal. Providing empirical learning curves or a brief theoretical discussion showing that both agents remain aligned with the global objective over time would significantly strengthen the methodology.

Empirically, the performance gains achieved by PromptPilot could be better contextualized. When compared to the recent single-agent RL baseline FM-PPO (Liu et al., CVPR 2025), the improvements appear relatively modest, such as a 0.4% increase in mIoU on VOC and a 0.3% increase in DSC on JSRT. Given the inherent architectural complexity of coordinating three separate Q-networks, it would be beneficial for the authors to dive deeper into specific failure cases of single-agent models where the MARL framework demonstrates a clear and undeniable advantage. Furthermore, to ensure statistical rigor, I suggest reporting the mean and standard deviation across multiple RL initialization seeds rather than just variations in the reference images, as RL policies can be quite sensitive to seed variance. Including a "Random Manager" baseline in your ablation studies would also convincingly isolate the value added by your Shapley-guided arbitration logic.

---

> ### Author Rebuttal · Authors · 2026-03-29
>
> Thank you for these comments.
>
> ##  To W1
>
> In Promptpilot, **the goal is the actual marginal effect of the selected action under the current state, not shapley value of all points**. The decision process is sequential: at each step, the feature and physical agent each propose one candidate action, and manager selects only one of them. So, we use LOO under the shapley idea of marginal contribution. **The value change after removing the current action gives an estimate of how much that action helps the current prompt set. EMA is then used to smooth this estimate with historical contribution, so the manager is not impacted by fluctuation from single step**.
>
> The term “credit shadowing” is not an established term and looks like a made-up phrase, which mean is confusing. The issue of redundant prompts is less critical. **In practice, truly redundant points are already discouraged by internal reward. Such as, one semantically useful point has been selected, physical agent tends to propose another action with a larger physical distance, rather than a nearly overlapping one**. If this new point is still semantically consistent, that is exactly the desired behavior: **keeping feature consistency while enlarging spatial coverage**.
>
> ## To W2
>
> The graph is defined on target image patches. The cross image distance $|f_i^r - f_j^t|_2$ is used only for label propagation to initialize $V_p$ and $V_n$. After that, **both $M_f$ and $M_p$ are computed within the target image**. We use $L_2$ in feature space because **DINOv2 also uses euclidean distance for patch matching analysis in its paper**, which keeps the matching metric consistent. In physical space, it is suitable to measure pixel distance because prompt locations lie on the 2D plane.
>
> ## To W3
>
> Average time per image in prompt optimization on four medical datasets (seconds) is shown below. Our method is much more efficient because FM-PPO uses a Q-table, while ours is DQN-based.
>
> | Method | Train | Test |
> |---|---:|---:|
> | FM-PPO | 156.654 | 8.974 |
> | Ours | 8.504 | 0.711 |
>
> ## To W4
>
> The concern about Eq. 10 is based on a misunderstanding. Asymmetric reward is intentional, because **two agents are not to optimize the overall segmentation goal by themselves, but to optimize different sub-tasks in feature or physical space under the same state**, while the manager aligns the final decision with the segmentation goal. The **unselected agent can keep updating through its internal reward and provide better candidate actions later**.
>
> ## To W5
>
> VOC and JSRT are simple datasets, where the **strong baseline already performs well and the room for improvement is limited**. More important, PromptPilot shows **much larger gains on more challenging datasets in Table 1, including +9.4 on COCO, +4.4 on Kvasir, and +6.5 on GBM**. This is also **supported by the multi-reference results in Tables 5–8 and by noisy prompts in Table 9**, we achieves the best performance, lower variance, and remains effective under noisy prompts.
>
> Random manager is not valid. If it still uses DSC, it is the “manager + DSC, no shapley” setting already in Table 3. If it uses no DSC, then it is no longer a manager and nothing constrains action selection. It is the “single agent, without manager” already in Table 3.
>
> Our method is not sensitive. Across three seeds, the DSC remains very stable:
>
> | Seed | ISIC | Kvasir | GBM | JSRT |
> |---|---:|---:|---:|---:|
> | 0 | 78.7 | 49.5 | 72.5 | 89.7 |
> | 42 | 79.0 | 49.7 | 72.8 | 90.1 |
> | 123 | 78.8 | 49.4 | 72.4 | 89.6 |
>
> ## To Q1
>
> If permitted by the organizers, the title will be revised to avoid duplication.
>
> ## To Q2
>
> The term **“orthogonal” in our paper is used at the optimization level, not in the strict mathematical sense of statistical independence**. If “orthogonal” is too strong, “decoupled” will be used in the final version. Our point is that **semantic consistency and spatial coverage are two competing objectives**: emphasizing **semantic consistency tends to concentrate prompts on highly discriminative local regions**, while improving **spatial coverage requires expanding beyond these regions and may introduce less pure feature responses**. This view is consistent with prior work in Bae et al. and Kim et al.. So these two objectives are not simply complementary, but can pull optimization in different directions.
>
> In our method, the EMA momentum factor is denoted by α, while the DQN discount factor is denoted by a. We will revise it in the final version to avoid ambiguity.
>
> Kim, Beomyoung, et al. *Discriminative Region Suppression for Weakly-Supervised Semantic Segmentation*
> Bae, Wonho, et al. *Rethinking Class Activation Mapping for Weakly Supervised Object Localization*
>
> ## To Q3
>
> PerSAM, PerSAM-F, and VRP-SAM are **not methods that perform segmentation through point prompts in SAM, and point are not available**. So point prompts were not overlaid on the images in Figure 1. Point overlay visualizations of other methods will be added in final appendix.

---

> > ### Author Rebuttal · Reviewer_9ucL · 2026-04-01
> >
> > Thank you for the rebuttal and the additional clarifications regarding random seeds and terminology. However, after carefully reviewing your responses, several core theoretical and empirical concerns remain inadequately addressed.
> >
> > **1. The Mathematical Discrepancy Between LOO and Shapley Value**
> >
> > Regarding your fixation on the term "credit shadowing": I used this as a descriptive phrase to illustrate a well-documented mathematical phenomenon in cooperative game theory. Specifically, when two features (or prompts) are highly redundant, Leave-One-Out (LOO) attribution assigns zero marginal contribution to both, whereas the exact Shapley value distributes the credit equally. While you may take issue with the phrasing, dismissing the critique based on semantics does not erase the fundamental mathematical limitation of LOO. Relying on the physical agent's distance penalty is a practical engineering workaround, but it does not bridge the theoretical gap between your LOO implementation and true Shapley values. A rigorous manuscript must explicitly acknowledge and discuss this theoretical compromise, which your rebuttal refused to do.
> >
> > **2. Missing Computational Overhead Metrics (FLOPs)**
> >
> > In my initial review, I explicitly requested a comparison of the FLOPs required for your inference-time optimization. Your rebuttal provided the inference time (0.711s) but completely ignored the request for FLOPs. Given that LOO evaluation requires multiple forward passes through the SAM decoder for every prompt removal, reporting FLOPs is non-negotiable for a transparent evaluation of the true computational burden. Time metrics alone are hardware-dependent and insufficient for theoretical completeness.
> >
> > **3. Lack of Empirical Proof for MARL Stability**
> >
> > I raised a specific concern regarding policy divergence due to the asymmetric reward mechanism (Equation 10) and explicitly requested empirical learning curves to demonstrate that the unselected agent remains aligned with the global objective over time. Your rebuttal merely reiterated your design motivation ("this is intentional") without providing the requested empirical evidence. In MARL, theoretical assurances without empirical convergence proofs (learning curves) are inadequate to dispel concerns about training stability.
> >
> > Given the missing empirical evidence (FLOPs, learning curves) and the lack of theoretical depth in addressing the LOO approximation, the manuscript does not yet meet the rigorous standards of the conference. Consequently, I will maintain my original score.

---

> > > ### Author Response · Authors · 2026-04-01
> > >
> > > **This is the last round of response available to us under this process, and your irresponsible attitude leaves us no choice but to answer in these terms**.
> > >
> > > Your acknowledgement does not engage with our rebuttal in any concrete way. You say that “several core theoretical and empirical concerns remain inadequately addressed,” but **you never explain which point is still unresolved, which part of our response is supposedly wrong, or what theoretical issue actually remains**. Simply restating your conclusion without raising any concrete follow up question is irresponsible, and it is not how the ICML discussion process is supposed to work.
> > >
> > > **We answered your review point by point**, including the shapley issue, the graph formulation, the inference cost, the asymmetric reward design, and the seed stability. Repeating a general conclusion without engaging any of the actual content of our response is not a serious discussion.
> > >
> > > In particular, **“credit shadowing” is an AI-generated fabricated term, not a recognized technical concept in this context**, yet you used it to support a central criticism of our method. **You did not respond to this issue at all after we pointed it out. A criticism built on a fabricated term should not be presented as if it had a legitimate theoretical basis**.
> > >
> > > ICML does not only provide a rebuttal stage. **It also provides a further discussion stage precisely so that reviewers can follow up on points they still find unclear, and authors can respond to those specific concerns**. The purpose of this stage is to clear up misunderstandings, incorrect assumptions, and unresolved questions. **If you still reject our response, then you should say clearly what you think remains unresolved. You did not do that**. Instead, you irresponsibly claimed that “several core theoretical and empirical concerns remain inadequately addressed”.

---

### Decision · Program_Chairs · 2026-04-30

**Decision:**

Accept (regular)

**Comment:**

The paper addresses a highly relevant and challenging problem in a novel way. The experimental evaluation is rigorous, and the results are well supported. However, I suggest that the authors follow reviewers' suggestions in:

- Reframing of the LOO mechanism without overclaiming Shapley value approximation. I suggest adding the additional discussion on the Shapley-based attribution mechanism

- To revise the abstract and the introduction to better align with the scope of the paper